# Direct observation of electron transfer in solids through X-ray crystallography

**Daiji Ogata[1], Shota Koide[1], Hiroyuki Kishi ®[1] & Junpei Yuasa ®[1] ✉**

Nanoscale electron transfer (ET) in solids is fundamental to the design of multifunctional nanomaterials, yet its process is not fully understood. Herein, through X-ray crystallography, we directly observe solid-state ET via a crystal-to-crystal process. We first demonstrate the creation of a robust and flexible electron acceptor/acceptor (A/A) double-wall nanotube crystal ($[(Zn^{2+})_4(L_A)_4(L_{A=O})_4]_n$) with a large window (0.90 nm × 0.92 nm) through the one-dimensional porous crystallization of heteroleptic $Zn_4$ metallocycles (($Zn^{2+})_4(L_A)_4(L_{A=O})_4$) with two different acceptor ligands (2,7-bis((1-ethyl-1H-imidazol-2-yl)ethynyl)acridine ($L_A$) and 2,7-bis((1-ethyl-1H-imidazol-2-yl)ethynyl)acridin-9(10H)-one ($L_{A=O}$)) in a slow-oxidation-associated crystallization procedure. We then achieve the bottom-up construction of the electron donor incorporated-A/A nanotube crystal ($[(D)_2 \subset (Zn^{2+})_4(L_A)_4(L_{A=O})_4]_n$) through the subsequent absorption of electron donor guests (D = tetrathiafulvalene (TTF) and ferrocene (Fc)). Finally, we remove electrons from the electron donor guests inside the nanotube crystal through facile ET in the solid state to accumulate holes inside the nanotube crystal ($[(D^{•+})_2 \subset (Zn^{2+})_4(L_A)_4(L_{A=O})_4]_n$), where the solid-state ET process ($D - e^- \rightarrow D^{•+}$) is thus observed directly by X-ray crystallography.

Nanoscale electron transfer (ET) in solids is fundamental to the design of multifunctional nanomaterials[1,2], yet its process is insufficiently understood[3,4]. Among nanomaterials, nanotubes are a fascinating nanomaterial owing to their unique structures[5–7], which offer a variety of unique electronic states through electron and hole injection to accumulate electrons and holes in the nanotubes[8,9]. Despite their fascinating ET properties, carbon-based nanotube materials are difficult to control in terms of their size and shape owing to their extreme synthesis conditions, such as high temperatures. Conversely, a viable strategy for fabricating well-defined nanotubes with high tunability is a bottom-up construction of non-covalent nanotubes through the infinite one-dimensional (1D) columnar organization of organic[10–21] or metallic macrocycles[22–28], which sometimes offers crystalline-form nanotube materials. However, non-covalent nanotube crystals are not robust enough to be subjected to electron and hole injection, as these ET events generate active radical sites on the building constituents and can break the non-covalent interactions and destroy the crystalline state. Therefore, if one could succeed in the bottom-up construction of robust and flexible non-covalent nanotube crystals such that hole or

electron accumulation occurs through a crystal-to-crystal process, the direct observation of thermal ET in solids through X-ray crystallography would be possible. Direct ET observation through X-ray crystallography has greatly benefited various fields in science, yet its success has been limited in the observation of the photo-induced charge separation process without stoichiometric changes[29–31].

Herein, we demonstrate a direct observation of the solid-state ET based on X-ray crystallography through a crystal-to-crystal process of hole accumulation in electron-acceptor/acceptor (A/A) multi-wall nanotube crystal incorporating electron-donor guests (D = tetrathiafulvalene (TTF) and ferrocene (Fc)). First, we present the creation of an A/A double-wall nanotube crystal using a novel supramolecular crystallization method involving slow oxidation[32] associated crystallization, by which controlled crystallization of heteroleptic[33] $Zn_4$ metallocycles having a double-wall structure into a 1D porous framework is achieved (Fig. 1a–c). Owing to its unique double-wall structure with large windows (0.90 nm × 0.92 nm), the resulting $Zn_{4n}$ double-wall nanotube crystal ($[(Zn^{2+})_4(L_A)_4(L_{A=O})_4]_n$) is robust and flexible enough to maintain its crystalline state upon ET oxidation

[1]Department of Applied Chemistry, Tokyo University of Science, 1-3 Kagurazaka, Shinjuku-ku, Tokyo 162-8601, Japan. ✉e-mail: yuasaj@rs.tus.ac.jp

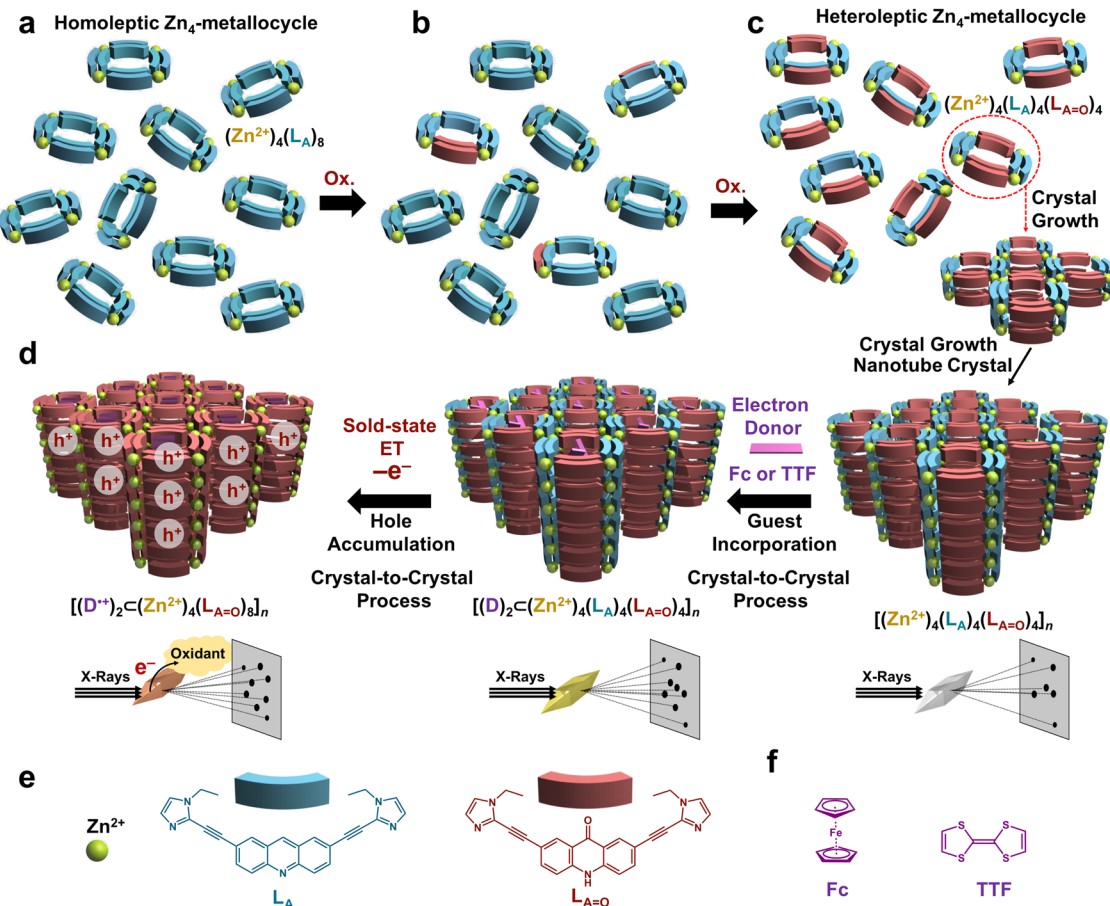

**Fig. 1 | Strategy for direct observation of thermal ET in solids through X-ray crystallography. a–c** Slow-oxidation-associated crystallization of a heteroleptic $Zn_4$ metallocycle $[(Zn^{2+})_4(L_A)_4(L_{A=O})_4]$: **a** The initial homoleptic $Zn_4$ macrocycle $[(Zn^{2+})_4(L_A)_8]$ is highly soluble in the crystallization solvent (acetonitrile/1,4-dioxane), and thus, no crystal growth can occur. **b** Slow oxidation of $(Zn^{2+})_4(L_A)_8$ by oxygen gradually yields heteroleptic $Zn_4$ metallocycles $((Zn^{2+})_4(L_A)_{(8-m)}(L_{A=O})_m)$ comprising $L_A$ and its corresponding oxidized acridone ligands $(L_{A=O})$ that have much lower solubility in the crystallization solvent. **c** Consequently, crystallization occurs gradually with an increasing $L_{A=O}/L_A$ ratio, where the $Zn_4$ heteroleptic metallocycle $((Zn^{2+})_4(L_A)_4(L_{A=O})_4)$ suitable for one-dimensional columnar stacking crystallizes preferentially over other competing metallocycles. **d** Direct observation of ET in solids by X-ray crystallography through hole accumulation in electron-donor-incorporated nanotube crystal by solid-state ET oxidation. **e** Building blocks of heteroleptic $Zn_4$ metallocycle. **f** Electron donors used in this study.

processes. The A/A double-wall nanotube crystal is capable of absorbing electron donors, whereas the crystal-to-crystal bottom-up construction of a double-wall nanotube crystal incorporating electron donor guests $([(D)_2 \subset (Zn^{2+})_4(L_A)_4(L_{A=O})_4]_n)$ is made possible by the subsequent absorption of D at the crystalline state (Fig. 1d)[34]. Electrons can be removed from the inner electron donors by facile ET in the solid state to accumulate holes inside the nanotube crystal $([(D)_2 \subset (Zn^{2+})_4(L_A)_4(L_{A=O})_4]_n - 2ne^- \rightarrow [(D^{•+})_2 \subset (Zn^{2+})_4(L_A)_4(L_{A=O})_4]_n)$, whereas the nanotube crystal maintains the crystalline state (Fig. 1d). Thus, surprisingly, the thermal ET process in solids $(D - e^- \rightarrow D^{•+})$ can be directly observed by X-ray crystallography.

## Results and discussion

### Electron-donor incorporated double-wall nanotube crystal

A new acridine ligand ($L_A$) acting as an electron acceptor was designed and synthesized in accordance with imidazole-based ditopic ligands developed in our supramolecular systems (Fig. 1e)[35–37]. Self-assembly of the homoleptic and heteroleptic double-wall $Zn_4$ metallocycles [respectively $(Zn^{2+})_4(L_A)_8$ and $(Zn^{2+})_4(L_A)_4(L_{A=O})_4$] in solution was investigated by adopting nuclear magnetic resonance and electrospray ionization mass spectroscopy (Supplementary Note 2; Supplementary Figs. 1–6).

Slow diffusion of 1,4-dioxane into an acetonitrile solution containing $(Zn^{2+})_4(L_A)_8$ gave crystals suitable for single-crystal X-ray diffraction analysis (Fig. 2d, Supplementary Notes 3 and 4; Supplementary

Figs. 7–13), which revealed 1D porous double-wall nanotubes $([(Zn^{2+})_4(L_A)_4(L_{A=O})_4]_n)$ formed by the infinite 1D columnar stacking of $(Zn^{2+})_4(L_A)_4(L_{A=O})_4$. The heteroleptic $Zn_4$ host frame $((Zn^{2+})_4(L_A)_4(L_{A=O})_4)$ comprised two acridine dimer stacking units and two acridone dimer stacking units, where each dimer stacking unit was twisted; specifically, M-helicity for the acridine dimers and P-helicity for the acridone dimers (Fig. 2d). Hence, the obtained double-wall nanotubes had chirality[38], where the enantiomers (i.e., $[MMPP-(Zn^{2+})_4(L_A)_4(L_{A=O})_4]_n$ and $[PPMM-(Zn^{2+})_4(L_A)_4(L_{A=O})_4]_n)$ were stacked with each other in the crystal packing structure (Supplementary Fig. 14).

In the present study, we used ferrocene (Fc) and tetrathiafulvalene (TTF) (Fig. 1f) as prototypical electron donors to assess the viability of the electron-donor incorporation method for the double-wall nanotube at the crystalline state (Supplementary Notes 5 and 6; Supplementary Figs. 15–20; Supplementary Table 1), wherein both Fc and TTF could enter the $Zn_4$ host frame window (0.90 nm × 0.92 nm; Supplementary Figs. 21 and 22). Soaking the double-wall nanotube crystals $([(Zn^{2+})_4(L_A)_4(L_{A=O})_4]_n)$ in acetonitrile/1,4-dioxane (1:2, v/v) containing Fc or TTF for 7 days resulted in crystal color changes (Fig. 2a), indicating absorption of the guest molecules[39–41]. However, the crystals maintained their crystalline states and no appreciable shape change was observed in the $Zn_4$ host frame after absorption of the electron donors (Supplementary Fig. 25). Electron density maps ($F_o$) from the X-ray diffraction analysis of the Fc-

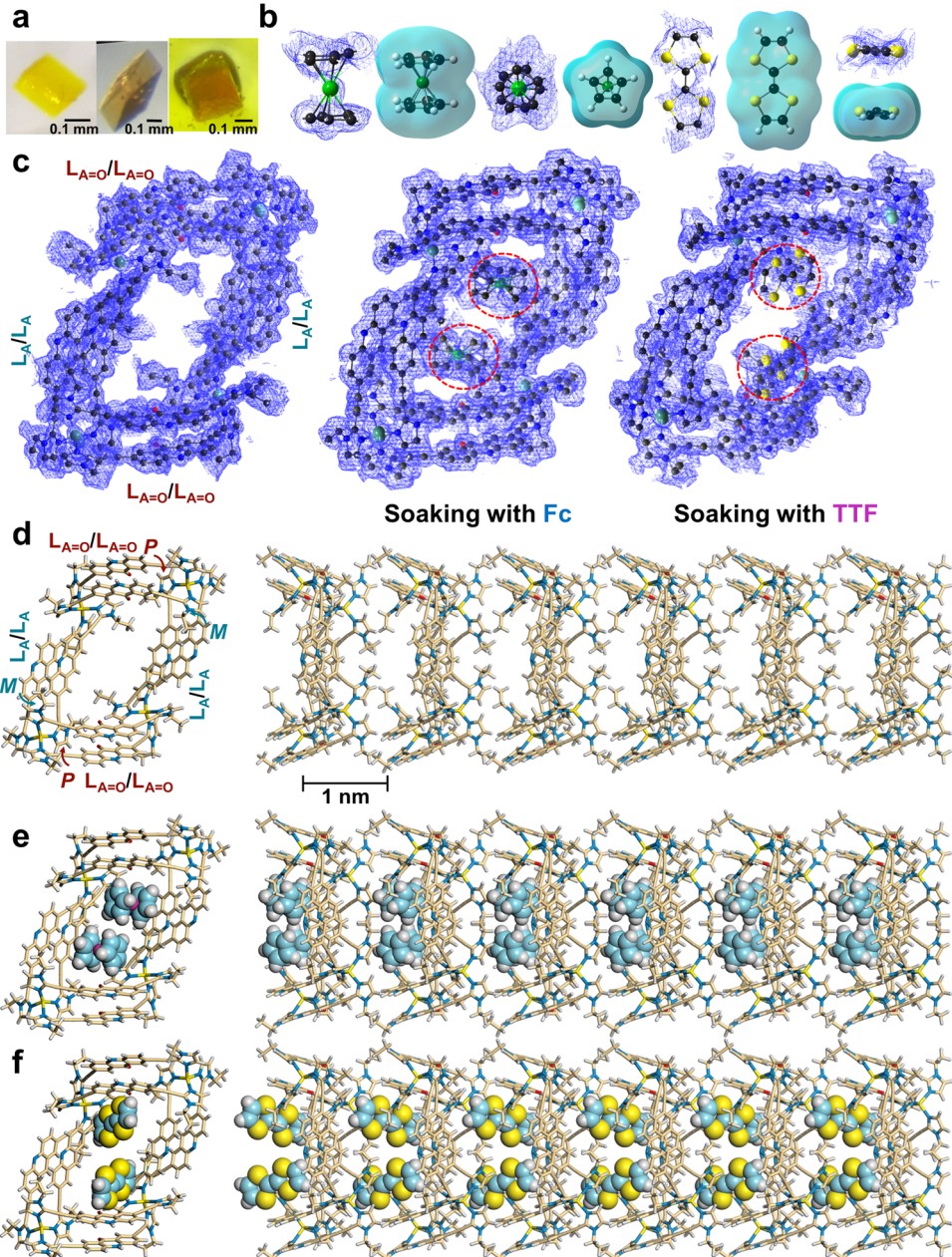

**Fig. 2 | One-dimensional organization of electron donors in the double-wall nanotube crystal. a** Photographs of double-wall nanotube crystals ($[(Zn^{2+})_4(L_A)_4(L_{A=O})_4]_n$) before (left) and after soaking in solutions containing Fc (middle) and TTF (right). **b** Newly observed electron densities ($F_o$) found in the Fc- and TTF-soaked crystals alongside the total electron densities of Fc and TTF structures as revealed by density functional theory [B3LYP/6-31 G + (d,p); LANL2DZ (Fe)]. Color code: green (Fe), yellow (S), and black (C). **c** Electron density map ($F_o$) of $(Zn^{2+})_4(L_A)_4(L_{A=O})_4$ before (left) and after soaking with Fc (middle) and TTF (right). Red circles show newly observed electron densities. The electron density ($F_o$) due to $OSO_2CF_3^-$ anions is omitted for clarity. **d–f** X-ray crystal structures of **d** $[(Zn^{2+})_4(L_A)_4(L_{A=O})_4]_n$, **e** $[(Fc)_2 \subset (Zn^{2+})_4(L_A)_4(L_{A=O})_4]_n$, and **f** $[(TTF)_2 \subset (Zn^{2+})_4(L_A)_4(L_{A=O})_4]_n$. One of the disordered structures is shown, and $OSO_2CF_3^-$ anions are omitted for clarity. Left figures: top views; right figures: side views.

and TTF-soaked crystals revealed new electron densities in the cavities of the Zn$_4$ host frame (Fig. 2c). The distributions of these new electron densities agreed well with those of Fc and TTF, as calculated adopting density functional theory (Fig. 2b). X-ray structural analysis of the Fc- and TTF-soaked crystals revealed that the absorbed Fc and TTF were aligned in one direction along the inner channel cavity (Fig. 2e, f; Supplementary Tables 2 and 3). A maximum of two Fc or TTF guests could be stored per Zn$_4$ host frame (Supplementary Figs. 23–26), and the chirality of the Zn$_4$ host frame was successfully transferred to the arrangement of the absorbed guests; i.e., *P*- and *M*-helicity was imparted by the *PPMM*- and *MMPP*-$(Zn^{2+})_4(L_A)_4(L_{A=O})_4$ host frames, respectively (Supplementary Fig. 27).

## Direct observation of thermal ET in solids through X-ray crystallography

Then, we investigated the direct observation of the thermal ET in solids using the above TTF- and Fc-incorporated double-wall nanotube crystals through X-ray crystallography. TTF and Fc have low one-electron oxidation potential and are suitable for facial ET oxidation by $[Fe(H_2O)_6](ClO_4)_3$ (Supplementary Fig. 28)[42,43]. Upon surface contact with the $[Fe(H_2O)_6](ClO_4)_3$ solids, there are gradual changes from the yellow color of the TTF- and Fc-incorporated nanotube crystals to the dark brown color of TTF radical cation (TTF$^{•+}$) and the dark color of ferrocenium ion (Fc$^+$), respectively (Fig. 3a, b; see Supplementary Movies 1 and 2). Furthermore, the

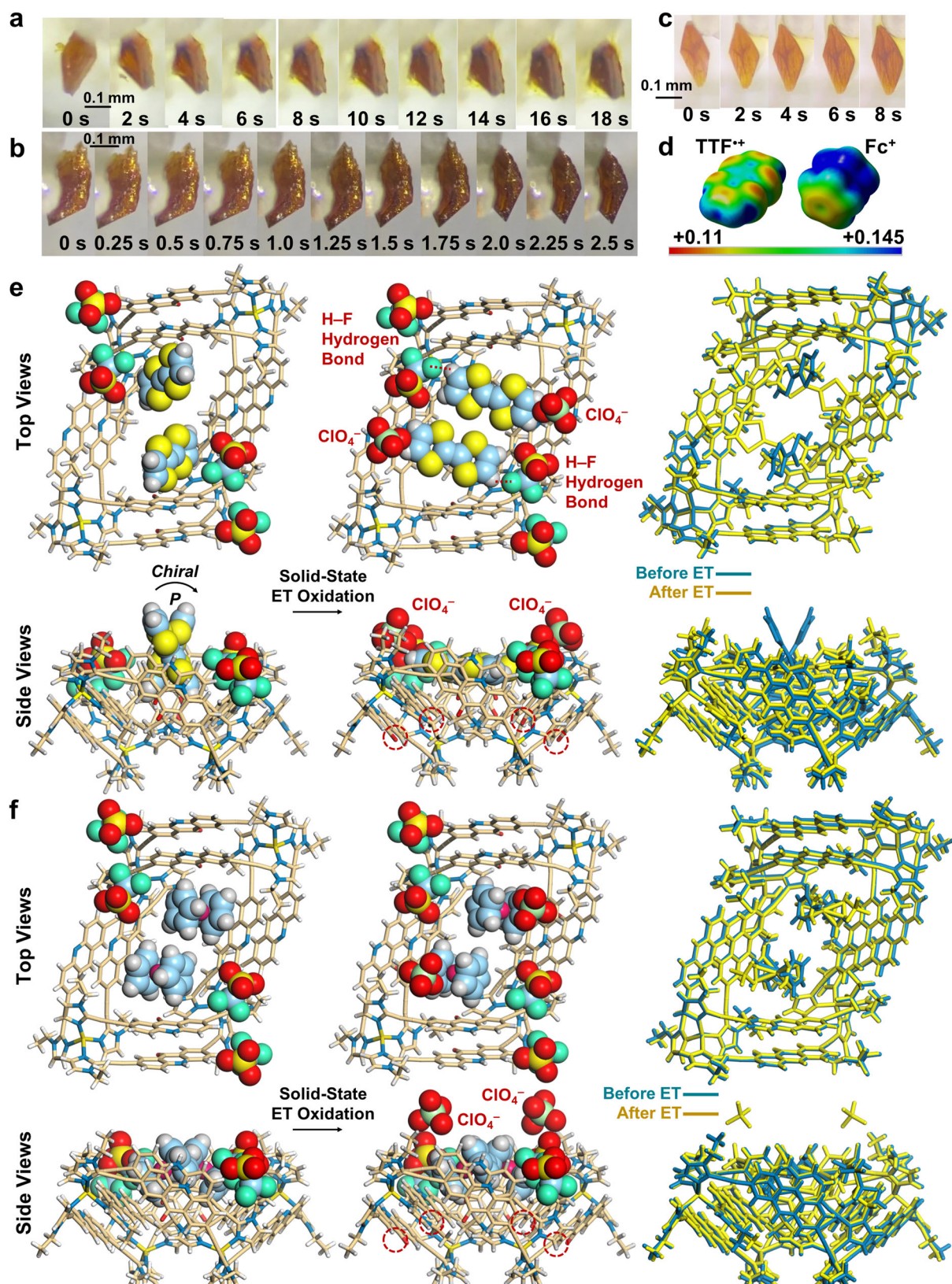

**Fig. 3 | Direct observation of thermal ET in solids through X-ray crystallography. a–c** Crystal photographs of **a** $[(TTF)_2 \subset (Zn^{2+})_4(L_A)_4(L_{A=O})_4]_n$ **b** $[(Fc)_2 \subset (Zn^{2+})_4(L_A)_4(L_{A=O})_4]_n$, and **c** $[(Zn^{2+})_4(L_A)_4(L_{A=O})_4]_n$ upon surface contact with the $[Fe(H_2O)_6](ClO_4)_3$ solid. **d** Electrostatic potential maps of TTF$^{\bullet+}$ and Fc$^+$ obtained with DFT [B3LYP/6-31 G + (d,p); LANL2DZ (Fe)]. **e**, **f** X-ray crystal structures of **e** $[(TTF)_2 \subset (Zn^{2+})_4(L_A)_4(L_{A=O})_4]_n$ and **f** $[(Fc)_2 \subset (Zn^{2+})_4(L_A)_4(L_{A=O})_4]_n$ before and after ET oxidation with $[Fe(H_2O)_6](ClO_4)_3$. One of the disordered structures was shown for clarity. Dashed circles indicate newly appeared oxygen atoms after the solid-state ET oxidation.

TTF-incorporated nanotube crystals exhibited an intense electron spin resonance signal due to TTF⁺⁺ after the ET oxidation (Supplementary Fig. 29). It is noteworthy that such remarkable color change of the crystal could not be observed for the electron-donor unoccupied nanotube crystal (i.e., $[(Zn^{2+})_4(L_A)_4(L_{A=O})_4]_n$) upon surface contact with the $[Fe(H_2O)_6](ClO_4)_3$ solid (Fig. 3c; see Supplementary Movie 3). Thus, the color changes observed for the TTF- and Fc-incorporated nanotube crystals (Fig. 3a, b) are attributed to the ET oxidation of the inner donor guests to accumulate holes inside the nanotube (Fig. 1d). Surprisingly, both TTF- and Fc-incorporated nanotube crystals maintained crystalline states during the ET oxidation (Supplementary Fig. 30), and we thus performed the X-ray structural analysis of the nanotube crystals after the ET oxidation. The X-ray structural analysis revealed that the acridine ligands of the host frame were almost fully oxidized to acridone in the generation of a homoleptic $Zn_4$ host frame $[(Zn^{2+})_4(L_A)_4(L_{A=O})_4 \rightarrow (Zn^{2+})_4(L_{A=O})_8]$ after the ET oxidation (Fig. 3e, f; Supplementary Figs. 31–35; Supplementary Tables 4–6), while no appreciable shape change (e.g., Zn-Zn distances) was observed in the $Zn_4$ host frame (Supplementary Figs. 36–38). Moreover, two $ClO_4^-$ newly appeared close to the inner TTF and Fc guests (Fig. 3e, f; Supplementary Figs. 39 and 40). The additional $ClO_4^-$ molecules originated from the counter anion of $[Fe(H_2O)_6]$ $(ClO_4)_3$ used as the oxidant, which compensated the additional positive charge in the nanotube crystal generated by the ET oxidation. In addition, after the ET oxidation, the two inner TTF guests moved toward the closest $OTf^-$ molecules to generate hydrogen bonds between the terminal hydrogen atoms of TTF and the F atoms of the $OTf^-$ molecules ($d_{H–F} = 2.572$ Å, Fig. 3e and Supplementary Fig. 41; see Supplementary Movie 4). An electrostatic potential map of TTF⁺⁺ suggests that the terminal hydrogen atoms were most positively charged (Fig. 3d), which explains the observed movement of the inner TTF molecules by the ET oxidation. Intriguingly, the positions of the two inner TTF molecules changed from the twisted conformation to the parallel conformation after the ET oxidation ($\theta = 66.4° \rightarrow 11.8°$); therefore, the chirality was almost lost in the arrangement of the two TTF guests (Fig. 3e). In contrast, only a slight positional change ($\theta = 60.7° \rightarrow 57.4°$) was observed for the Fc guests after the ET oxidation (Fig. 3f; see Supplementary Movie 5). Fc⁺ has no acidic hydrogen atom to form a hydrogen bond with the F atom of the $OTf^-$ molecule (Fig. 3d), thus no significant positional change can be expected before and after the ET oxidation of the inner Fc guests. Conversely, the two $ClO_4^-$ molecules that appeared after the ET oxidation were located at the more inner regions of the $Zn_4$-host frame than that of the TTF-incorporated nanotube crystals (Fig. 3f vs. e), which should be attributed to the high positive charge of the central $Fe^{3+}$ of the inner Fc⁺ guests (Fig. 3d). Although the detailed solid-state ET mechanism in this system is still unclear at present, the tubular void with a large window (0.90 nm × 0.92 nm) should play an important role in delivering the additional $ClO_4^-$ and holes generated at the surface into the depths of the tube (Supplementary Note 7; Supplementary Figs. 43–45). This is fundamentally different from the crystal-to-crystal transition of the redox-responsive metal-organic-frameworks (MOF) (Supplementary Fig. 46)[44–46].

Since the above X-ray crystallography method can directly determine the initial and the final structures of the solid-state ET process (Fig. 3e, f), this opens a way for direct determination of the reorganization energy ($\lambda$) of the thermal ET in solids within the framework of the Marcus-Hush two-state model (Supplementary Note 8; Supplementary Figs. 42 and 43)[47–49]. The present analysis revealed unusually large $\lambda$ values for the solid-state ET oxidation of TTF and Fc inside the nanotube (1.36 and 2.23 eV, respectively). Such large $\lambda$ values are mostly attributed to the incoming $ClO_4^-$ molecules to compensate for the additional

positive charge on the electron donor molecules after the ET oxidation. Such large $\lambda$ normally decelerates the ET rate, and thus the findings give a convincing explanation for the fact that the solid-state ET system is still quite rare. An efficient ET in solids should be rather rigid showing only small composition changes in response to the oxidation/reduction, like typical ET of biological cofactors[50].

This result is a striking example of the direct observation of a thermal ET process in solids through X-ray crystallography and demonstrates that the scope of the proposed method can be expanded to cover ET events in nanomaterials. A variety of solid-state ET processes can be investigated in a similar way.

## Methods

### General methods
¹H and ¹³C NMR spectra were measured with JEOL JNM-ECZ400S, JNM-ECA500 and Bruker AVANCE NEO 400. UV-Vis absorption spectra were recorded by a JASCO V-660 at ambient temperature. High-resolution electrospray ionization (HR-ESI) mass spectra were measured with mass spectrometers X500R QTOF (Sciex, MA, USA). DFT studies were performed with the GAUSSIAN '09 package. The atomic coordinate files after DFT calculations are included as Supplementary Data files 1–12.

### Synthesis and characterization
**2,7-Bis((1-ethyl-1H-imidazol-2-yl)ethynyl)acridine (L$_A$).** To a two-necked flask, CuI (176 mg, 0.928 mmol), 2,7-bis(ethynyl-trimethylsilane)acridine (3.45 g, 9.28 mmol), 1-ethyl-2-iodo-1H-imidazole (5.15 g, 23.2 mmol), THF (80 mL), and triethylamine (60 mL) were added. After the solution was degassed by bubbling with Ar gas for 30 min, tetrabutylammonium fluoride (in tetrahydrofuran 1 mol/L, 27.8 mL) and Pd(PPh₃)₄ (1.07 g, 0.928 mmol) were added to the flask. Then, the reaction mixture was refluxed under an Ar atmosphere for 24 h. The filtrate was extracted with chloroform. The organic phase was consecutively washed with water and then brine. It was dried over Na₂SO₄, filtered and the solvent was evaporated, the crude product was subjected to column chromatography on silica gel (chloroform/methanol = 9/1) and purified by GPC with chloroform to afford a yellow solid (1.26 g, 32.7%). ¹H NMR (500 MHz, 298 K, CD₃CN) $\delta$ 8.94 (s, 1H), 8.41 (s, 2H), 8.18 (d, $J = 8.9$ Hz, 2H), 7.93 (dd, $J = 8.9$, 1.7 Hz, 2H), 7.22 (s, 2H), 7.06 (s, 2H), 4.24 (q, $J = 7.2$ Hz, 4H), 1.48 (t, $J = 7.2$ Hz, 6H). ¹³C NMR (125 MHz, 298 K, CDCl₃) $\delta$ 148.59, 135.79, 132.55, 131.94, 131.12, 130.10, 129.74, 126.27, 119.88, 119.78, 92.18, 80.63, 41.95, 16.07 ppm. HRMS (ESI): $m/z$ calcd for $[C_{27}H_{21}N_5 + H]^+$: 416.18752; found: 416.18681. Synthesis and characterization details can be found in the Supplementary Note 1.

### X-ray crystallographic analysis
X-ray diffraction data were collected on Bruker-AXS· D8 QUEST using microfocus Mo$K\alpha$ radiation ($\lambda = 0.71073$ Å) equipped with a CCD detector. All data collection strategies were performed at 90 K using cold nitrogen streams. X-ray crystallographic analysis details can be found in the Supplementary Note 1. Cif files and checkcif files are included as Supplementary Data files 13-24.

### Reporting summary
Further information on research design is available in the Nature Portfolio Reporting Summary linked to this article.

## Data availability
The X-ray data have been deposited at the Cambridge Crystallographic Data Centre (CCDC) under reference numbers 2262473–2262475; 2298311–2298313. All data are available in the main text or the supplementary materials. Correspondence and requests for materials should be addressed to Junpei Yuasa (e-mail: yuasaj@rs.tus.ac.jp).

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

## Acknowledgements

We thank Dr. Jay Freeman at Edanz (https://jp.edanz.com/ac) for editing a draft of this manuscript. This work was partly supported by JSPS KAKENHI under grant numbers JP23H01941 (J.Y.) and JP23H0397 (J.Y.) (Scientific Research on Innovative Areas, Dynamic Exciton) and JP21J20598 (D.O.).

## Author contributions

D.O., H.K. and K.S. performed the synthesis and characterization of materials. D.O. and H.K. analyzed the experimental data, including the X-ray crystal structures. J.Y. designed the study, analyzed the experimental data, and wrote the manuscript.

## Competing interests

The authors declare no competing interests.
