## [Peer Review File · Nature Communications]

Direct observation of electron transfer in solids through X-ray crystallographyReviewer #2 (Remarks to the Author):

The paper details the synthesis of an unusually robust noncovalent nanotube framework, which maintains single crystallinity even upon guest absorption and subsequent oxidation. While the science presented is nicely characterized, I disagree with the claim that the direct observation of solid-state electron transfer by X-ray crystallography has limited precedent. For example, several examples of single crystal-to-single crystal redox transformations in metal-organic frameworks are highlighted below:

- <https://pubs.acs.org/doi/10.1021/ja0466715>
- <https://pubs.acs.org/doi/full/10.1021/acs.inorgchem.8b03299>
- <https://www.nature.com/articles/s41467-017-02256-y>

As these prior reports unfortunately diminish the paper's novelty, I do not believe this paper has sufficient impact to be published in Nature Communications.

Reviewer #3 (Remarks to the Author):

The authors report new tube-like host compound that shows interesting oxidation process in solids. It can be accepted after revising the following things.

- 1) This paper describes direct electron transfer (ET) observation in solids through X-ray crystallography. But, in text, I found some expression, i.e., "thermal ET process in solids". When the behavior was occurred through X-ray irradiation, the process is related to light, not some heat. Please mention it.
- 2) In Fig. 1a, $(Zn^{2+})_4(LA)_8$ host compound changed to $(Zn^{2+})_4(LA)_4(LA=O)_4$ by oxidation. In this case, what is the source of oxygen atom? (i.e., H_2O). And what is the reason of the oxidation from LA to LA=O? (i.e., driving force for the oxidation). Moreover, in $(Zn^{2+})_4(LA)_4(LA=O)_4$ host compound, the other four LA species did not changed to LA=O. Any possible reason for it?
- 3) $[Fe(H_2O)_6](ClO_4)_3$ is used as oxidation agent. What is the benefit for this reaction? If it is used any oxidation reagents without oxygen atoms, $(Zn^{2+})_4(LA)_8$ host compound can maybe maintain as original form without any oxidation. Please check and mention it.
- 4) $(Zn^{2+})_4(LA)_4(LA=O)_4$ host compound can include an Fc or TTF, separately. What happens when combine both guest molecules simultaneously? (comparing to Figure S31)
- 5) In Figure S23A and S23C, the currents of TTF and TTF-soaking crystal show some thickness, not just line (comparing to Figure S26). Is there any reason?

Reviewer #5 (Remarks to the Author):

Nanoscale electron transfer (ET) in the solid state is of crucial importance in fabricating functional nanomaterials and molecular machines. The present work is nicely designed and carried out. The authors have been able to show ET in the solid state directly via X-ray crystallography. Detail methodology has been provided that should help in reproducing this and similar studies in future. I recommend that this work may be accepted in Nature Communications.

Tokyo University of Science

Department of Applied Chemistry
1-3 Kagurazaka, Shinjuku, Tokyo 162-8601, Japan

E-mail: yuasaj@rs.tus.ac.jp

March 27, 2024

Manuscript number: NCOMMS-23-53731-T

Title: Direct observation of electron transfer in solids through X-ray crystallography

Authors: Daiji Ogata, Shota Koide, Hiroyuki Kishi, Junpei Yuasa*

Dear Reviewers:

We appreciate the reviewer's efforts and professional advice to improve our manuscript. The professional suggestions from the reviewers (reviewer 2, 3, and 5) are really useful to improve our manuscript. We have made changes suggested by the reviewers and have responded to each of the points as indicated below.

Reviewer 2:

Reviewer's Comments: The paper details the synthesis of an unusually robust noncovalent nanotube framework, which maintains single crystallinity even upon guest absorption and subsequent oxidation. While the science presented is nicely characterized, I disagree with the claim that the direct observation of solid-state electron transfer by X-ray crystallography has limited precedent. For example, several examples of single crystal-to-single crystal redox transformations in metal-organic frameworks are highlighted below:

- <https://pubs.acs.org/doi/10.1021/ja0466715>
- <https://pubs.acs.org/doi/full/10.1021/acs.inorgchem.8b03299>
- <https://www.nature.com/articles/s41467-017-02256-y>

As these prior reports unfortunately diminish the paper's novelty, I do not believe this paper has sufficient impact to be published in Nature Communications.

Answer: *First of all, we appreciate the reviewer's efforts on reviewing our manuscript.* We also appreciate the reviewer's kind suggestion with regard to the previous reports on the redox-responsive metal-organic-frameworks (MOF) and their crystal-to-crystal transition. We have read these important previous reports carefully and found that these works contained a high level of chemistry. These manuscripts have been cited in the revised manuscript (ref. 44-46). Conversely, our present work is *fundamentally different* from the prior reports (ref. 44-46) in the following points:

(i) The previously reported redox-driven crystal-to-crystal transition of MOF relies on the electron-transfer (ET) oxidation of the *host frame* (i.e., oxidation of the TTF-derivatized host ligands (ref. 45 and 46) or charge up of the Ni-contained ligands (ref. 44)) [Figure S46a]. Conversely, our manuscript reports the crystal-to-crystal process of the electron-transfer

oxidation of the *guests* (non-derivatized TTF and Fc) *non-covalently incorporated* into the nanotube channels (Figure S46b). Thus, our non-derivatized free system is highly versatile for direct observation of the solid-state electron-transfer process of the electron donor molecule itself by X-ray crystallography.

(ii) In addition, the previous reports used solution (or vapor) of iodine (I_2) to oxidize the (MOF) crystal. In contrast, we used the $[Fe(H_2O)_6](ClO_4)_3$ solid to oxidize the nanotube crystal.

(iii) Furthermore, in the previous reports (ref. 44-46), the oxidant of I_2 (dissolved in solution) was most probably directly incorporated into the host-frame channels. Subsequently, electron transfer occurred directly between the host-frame and the incorporated oxidant (I_2) inside the channels (Figure S46a). Thus, after the electron transfer, the incorporated I_2 molecules reduced to I_3^- (worked as counter anion) were kept included in the channels (Figure S46a). Conversely, in our system, the electron-transfer oxidation of the inner guest was occurred outside the host-frame channels, i.e., the crystal surface contacted with the oxidant ($[Fe(H_2O)_6](ClO_4)_3$ solid). The size of $[Fe(H_2O)_6]^{3+}$ is too large to pass the window of the $(Zn^{2+})_4(L_A)_4(L_{A=O})_4$ host frame (0.90 nm \times 0.92 nm) [Figure S44a], and the incorporation of the positively charged $[Fe(H_2O)_6]^{3+}$ into the positively charged $[(Zn)_4(L_A)_4(L_{A=O})_4]^{8+}$ host frame is unlikely to occur. Subsequent hole transfer and the incorporation of ClO_4^- into the nanotube channel resulted in oxidation of the whole range of the guest molecules inside the nanotube crystal (Figure S46b). Hence, the outer $[Fe(H_2O)_6]^{3+}$ reduced to $[Fe(H_2O)_6]^{2+}$ was kept outside the nanotube crystal, where the additional counter anion of ClO_4^- was incorporated into the host-frame channels after the solid-state electron transfer (Figure S46b). The size of ClO_4^- is small enough to pass the host-frame window size (0.90 nm \times 0.92 nm) [Figure S44b], where the incorporation of the negatively charged ClO_4^- into the positively charged $[(Zn)_4(L_{A=O})_8]^{8+}$ host frame is likely to occur. Indeed, the additional ClO_4^- molecules were appeared inside the nanotube channel after the electron-donor oxidation, which was directly determined by X-ray crystallography (Fig. 3e).

Thus, the previous reports (ref. 44-46) *do not diminish the paper's novelty*, and the other reviewers (reviewer 3 and 5) admitted the novelty of our work suitable for publication in *Nature Communications* (please see the reviewers' comments).

Once again, we appreciate the reviewer's efforts and professional advice to improve our manuscript.

Figure S44. Comparison between window size of the host $(Zn^{2+})_4(L_A)_4(L_{A=O})_4$ frame and (a) $[Fe(H_2O)_6]^{3+}$ and (b) ClO_4^- . Here, the X-ray crystal structure of $(Zn^{2+})_4(L_A)_4(L_{A=O})_4$ was used as the structures of the host frame, and DFT-optimized structures ([CAM-B3LYP/6-31G(d) [H O Cl]; LANL2DZ (Fe)]) were used as the guest structures.

Figure S46. Schematic representation for (a) redox-stimuli responsive MOF and (b) solid-state oxidation of $[(D)_2C(Zn^{2+})_4(L_A)_4(L_{A=O})_4]_n$ crystals.

Reviewer 3:

Reviewer's Comments: The authors report new tube-like host compound that shows interesting oxidation process in solids. It can be accepted after revising the following things.

Answer: *We appreciate the reviewer's high evaluation on our work "It can be accepted after revising the following things".*

1) This paper describes direct electron transfer (ET) observation in solids through X-ray crystallography. But, in text, I found some expression, i.e., "thermal ET process in solids". When the behavior was occurred through X-ray irradiation, the process is related to light, not some heat. Please mention it.

Answer: Thank you very much for your question. The ET oxidation of the nanotube crystal was completed by surface contact with the $[\text{Fe}(\text{H}_2\text{O})_6](\text{ClO}_4)_3$ solid prior to the X-ray irradiation (Fig. 3a and b; Supplementary Movie 1-3). In addition, the X-ray irradiation was performed at 90 K, hence no further reaction could not proceed during the X-ray irradiation. This point has been mentioned in the ESI (page 44 and 45).

2) In Fig. 1a, $(\text{Zn}^{2+})_4(\text{L}_A)_8$ host compound changed to $(\text{Zn}^{2+})_4(\text{L}_A)_4(\text{L}_{A=O})_4$ by oxidation. In this case, what is the source of oxygen atom? (i.e., H_2O). And what is the reason of the oxidation from L_A to $\text{L}_{A=O}$? (i.e., driving force for the oxidation). Moreover, in $(\text{Zn}^{2+})_4(\text{L}_A)_4(\text{L}_{A=O})_4$ host compound, the other four L_A species did not changed to $\text{L}_{A=O}$. Any possible reason for it?

Answer: *We appreciate the reviewer's professional question.* We consider that the source of oxygen atom in the present system is most probably oxygen in air. DFT calculation (CAM-B3LYP/6-31G+(d,p)) suggests that the oxidation reaction, $\text{L}_A + 1/2\text{O}_2 \rightarrow \text{L}_{A=O}$, is 53.3 kcal mol⁻¹ energetically down hill (while this is spin-forbidden oxygenation).

Then, we performed the negative control experiments in this revision (see below). 3 days standing of the acetonitrile/1,4-dioxane mixed solution (1/2, vol/vol) of L_A containing $\text{Zn}(\text{OTf})_2$ ($\text{OTf}^- = \text{OSO}_2\text{CF}_3^-$) **under air** resulted in conversion of L_A to $\text{L}_{A=O}$ partially (Figure S13a and d). Hence, the oxidation from L_A to $\text{L}_{A=O}$ proceeds slowly in the presence of oxygen in the solution phase. Conversely, almost no oxidized product ($\text{L}_{A=O}$) was formed in that performed **under the deaerated conditions** (Figure S13b and c). The negative result obtained **under the deaerated conditions** suggests that oxygen in air is the source of oxygen atom in the present system. These points have been mentioned in the ESI (page 16).

In addition, we planned to perform $^{18}\text{O}_2$ -labeled experiment for further verification. Unfortunately, commercially available $^{18}\text{O}_2$ -labeled oxygen (Sigma Aldrich) is currently not available in Japan.

As suggested by the reviewer, in the $(\text{Zn}^{2+})_4(\text{L}_A)_4(\text{L}_{A=O})_4$ host frame, the other four L_A ligands did not change to $\text{L}_{A=O}$. The oxidation of L_A to $\text{L}_{A=O}$ occurred mostly in the solution phase prior to the crystallization to form the double-wall nanotube, giving a mixture of $(\text{Zn}^{2+})_4(\text{L}_A)_m(\text{L}_{A=O})_{(8-m)}$ ($m = 0-8$) host compounds in the solution (Fig. 1a). Conversely,

oxidation of the $(\text{Zn}^{2+})_4(\text{L}_A)_4(\text{L}_{A=O})_4$ host frame could not be occurred in the crystalline state (without using the $[\text{Fe}(\text{H}_2\text{O})_6](\text{ClO}_4)_3$ solid). Among the $(\text{Zn}^{2+})_4(\text{L}_A)_m(\text{L}_{A=O})_{(8-m)}$ host compounds, the $(\text{Zn}^{2+})_4(\text{L}_A)_4(\text{L}_{A=O})_4$ host frame favorably crystallizes through the slow-oxidation-associated crystallization process as explained in Fig. 1a. In such a case, the composition of $(\text{Zn}^{2+})_4(\text{L}_A)_4(\text{L}_{A=O})_4$ is frozen in the crystal, and therefore the other four L_A ligands did not change to $\text{L}_{A=O}$ during the crystallization process. These points have been mentioned in the ESI (page 15) and the additional experimental results has been added in ESI (Figure S13).

Figure S13. (a,b) Schematic representation for the oxidation experiments of $(\text{Zn}^{2+})_4(\text{L}_A)_8$ in (a) under air and (b) under the deaerated conditions: Acetonitrile/1,4-dioxane mixed solutions (1/2, vol/vol) of L_A (2.0 mM) containing $\text{Zn}(\text{OTf})_2$ (2.0 mM) were standing for 3 days (a) under air and (b) under the deaerated conditions. Then, the solvent of the resulting solutions were removed by evaporation, and the crude products were dissolved in DMF- d_7 . (c,d) ^1H NMR spectra (in DMF- d_7) of the resulting samples obtained (c) under the deaerated conditions and (d) under air.

3) $[\text{Fe}(\text{H}_2\text{O})_6](\text{ClO}_4)_3$ is used as oxidation agent. What is the benefit for this reaction? If it is used any oxidation reagents without oxygen atoms, $(\text{Zn}^{2+})_4(\text{L}_A)_8$ host compound can maybe maintain as original form without any oxidation. Please check and mention it.

Answer: *Once again, we appreciate the reviewer's professional question.* Indeed, the solid-state oxidant available for solid-state electron-transfer reactions is quite limited. A $[\text{Fe}(\text{H}_2\text{O})_6](\text{ClO}_4)_3$ solid used in this study is one of the easy to use and mild solid-state oxidant. We also tested tris(4-bromophenyl)ammoniumyl hexachloroantimonate salts (magic blue) as a solid-state oxidant (without oxygen atoms) in the present system (vide infra). The crystallinity of the nanotube crystal $((\text{Zn}^{2+})_4(\text{L}_A)_4(\text{L}_{A=O})_4)_n$ was decreased after the surface contact with the magic

blue solid (Figure S45a) probably due to the radical nature of magic blue. Hence, we could not resolve the X-ray crystal structure due to the bad quality X-ray data. Conversely, the ratio between L_A and $L_{A=O}$ was determined by 1H NMR spectroscopy of the resulting crystals dissolved in $DMF-d_7$ (Figure S45b), indicating that the molar ratio ($[L_A]:[L_{A=O}] = 1:1$) almost unchanged even after the surface contact with the magic blue. Thus, the host frame ($(Zn^{2+})_4(L_A)_4(L_{A=O})_4$) could be maintain as original form by using magic blue solids. This point has been mentioned in ESI (page 42) and the additional experimental results has been added in ESI (Figure S45).

Figure S45. (a) Crystal photograph of the $[(Zn^{2+})_4(L_A)_4(L_{A=O})_4]_n$ crystal after surface contact with the tris(4-bromophenyl)ammoniumyl hexachloroantimonate solids. (b) Partial 1H NMR spectra (dissolving in $DMF-d_7$) of $[(Zn^{2+})_4(L_A)_4(L_{A=O})_4]_n$ crystals before (top) and after (bottom) the ET oxidation with tris(4-bromophenyl)ammoniumyl hexachloroantimonate solids. Blue and red circles denote the 1H NMR signals due to L_A and $L_{A=O}$, respectively.

4) $(Zn^{2+})_4(L_A)_4(L_{A=O})_4$ host compound can include an Fc or TTF, separately. What happens when combine both guest molecules simultaneously? (comparing to Figure S31)

Answer: Thank you very much for the reviewer's important question. We have performed the double guests (Fc or TTF) incorporation experiment as suggested by the reviewer in this revision. The nanotube crystal ($[(Zn^{2+})_4(L_A)_4(L_{A=O})_4]_n$) was immersed in the mixed solution of Fc or TTF in 7 days. The nanotube crystal color was changed from light yellow to dark brown (Figure S26), indicating the incorporation of the guest molecules into the nanotube crystal. Then, we analyzed the Fc/TTF-soaking crystal by X-ray crystallography, where new electron density was appeared inside the $(Zn^{2+})_4(L_A)_4(L_{A=O})_4$ host frame (Figure R1). However, the appeared new electron density could not be defined well probably due to statistical incorporation of Fc and TTF into the nanotube channel. This point has been mentioned ESI (page 29) and the additional experimental result (Figure S26) has been added in ESI.

Figure S26. Photograph of double-wall nanotube crystal ($[(\text{Zn}^{2+})_4(\text{LA})_4(\text{LA}=\text{O})_4]_n$) after soaking in solution containing Fc and TTF.

Figure R1. Electron density map (F_o) of $(\text{Zn}^{2+})_4(\text{LA})_4(\text{LA}=\text{O})_4$ before (left) and after soaking with the Fc/TTF mixed solution. Red circles show newly observed electron densities.

5) In Figure S23A and S23C, the currents of TTF and TTF-soaking crystal show some thickness, not just line (comparing to Figure S26). Is there any reason?

Answer: *We appreciate the reviewer's question.* Since the amount of the crystal sample (TTF-soaking crystal) is quite small, we used the specific electrochemical cell (SVC-2 Voltammetry cell/ Sample holder dia 9 mm ϕ) for the measurement of cyclic voltammetry (CV) for small amount of samples (Figure S23C). The thickness in CV curve was arising from the nature of the specific electrochemical cell. In order to fit the experimental conditions, we also used the specific electrochemical cell for the CV measurement of TTF for the calibration line (Figure S23A). This point has been described in the figure caption (Figure S24).

Once again, we appreciate the reviewer's efforts and professional advice on reviewing our manuscript.

Reviewer 5:

Reviewer's Comments: Nanoscale electron transfer (ET) in the solid state is of crucial importance in fabricating functional nanomaterials and molecular machines. The present work is nicely designed and carried out. The authors have been able to show ET in the solid state directly via X-ray crystallography. Detail methodology has been provided that should help in reproducing this and similar studies in future. I recommend that this work may be accepted in Nature Communications.

Answer: *We appreciate the reviewer's high evaluation on our work "I recommend that this work may be accepted in Nature Communications."*

Dr Junpei Yuasa

Department of Applied Chemistry
Tokyo University of Science
E-mail: yuasaj@rs.tus.ac.jp

Reviewer #2 (Remarks to the Author):

The authors' response and manuscript additions address my concerns.

Reviewer #3 (Remarks to the Author):

This revised version looks fine. Thus, the paper can be acceptable.

Tokyo University of Science

Department of Applied Chemistry
1-3 Kagurazaka, Shinjuku, Tokyo 162-8601, Japan

E-mail: yuasaj@rs.tus.ac.jp

April 20, 2024

Manuscript number: NCOMMS-23-53731-T

Title: Direct observation of electron transfer in solids through X-ray crystallography

Authors: Daiji Ogata, Shota Koide, Hiroyuki Kishi, Junpei Yuasa*

Dear Reviewers:

We appreciated complimentary comments from the reviewers (Reviewer #2: The authors' response and manuscript additions address my concerns.; Reviewer #23: This revised version looks fine. Thus, the paper can be acceptable.).

Prof. Dr Junpei Yuasa

Department of Applied Chemistry
Tokyo University of Science
E-mail: yuasaj@rs.tus.ac.jp